# Cre-Controlled CRISPR mutagenesis provides fast and easy conditional gene inactivation in zebrafish

Stefan Hans [1✉], Daniela Zöller[1], Juliane Hammer[1], Johanna Stucke[1], Sandra Spieß[1], Gokul Kesavan [1], Volker Kroehne [1], Juan Sebastian Eguiguren [1], Diana Ezhkova[1], Andreas Petzold[2], Andreas Dahl [2] & Michael Brand[1✉]

Conditional gene inactivation is a powerful tool to determine gene function when constitutive mutations result in detrimental effects. The most commonly used technique to achieve conditional gene inactivation employs the Cre/loxP system and its ability to delete DNA sequences flanked by two loxP sites. However, targeting a gene with two loxP sites is time and labor consuming. Here, we show Cre-Controlled CRISPR (3C) mutagenesis to circumvent these issues. 3C relies on gRNA and Cre-dependent Cas9-GFP expression from the same transgene. Exogenous or transgenic supply of Cre results in Cas9-GFP expression and subsequent mutagenesis of the gene of interest. The recombined cells become fluorescently visible enabling their isolation and subjection to various omics techniques. Hence, 3C mutagenesis provides a valuable alternative to the production of loxP-flanked alleles. It might even enable the conditional inactivation of multiple genes simultaneously and should be applicable to other model organisms amenable to single integration transgenesis.

[1] Center for Molecular and Cellular Bioengineering (CMCB), Center for Regenerative Therapies Dresden (CRTD), Technische Universität Dresden, Dresden, Germany. [2] Center for Molecular and Cellular Bioengineering (CMCB), DRESDEN-Concept Genome Center, Technische Universität Dresden, Dresden, Germany. ✉email: stefan.hans@tu-dresden.de; michael.brand@tu-dresden.de

Gene inactivation is the most powerful tool to determine gene function. Forward genetic screens in yeast, *C. elegans*, and *Drosophila* produced numerous mutants revealing genes and pathways controlling cell biology and development[1–3]. Similarly, saturation mutagenesis of the zebrafish genome yielded an array of mutations affecting all aspects of vertebrate development[4]. In mammalian species, reverse genetics is the dominant approach, building on precise gene targeting via homologous recombination in embryonic stem cells which has revolutionized the study of mammalian biology and human medicine[5]. Nowadays, designer nucleases like Zinc-finger nucleases, transcription activator-like effector nucleases (TALENs), and in particular clustered regulatory interspaced short palindromic repeat (CRISPR)/Cas9 are used for a broad range of tailor-made genomic modifications in almost all model organisms[6]. Designer nucleases enable targeted DNA double strand breaks that stimulate repair mechanisms which can subsequently be exploited for the generation of knock-out and knock-in alleles[7]. Knock-outs are achieved by error-prone nonhomologous

end joining DNA repair introducing small frameshift insertion–deletions (indels) in the coding sequence and the usage of resulting premature stop codons[8]. Knock-in generation employs either homology-directed repair for the precise integrations of small-sized cargo or homology-independent double strand break repair for the insertion of larger DNA cassettes[9–11].

Although mutations are key to understand gene function, the above-mentioned constitutive or germline gene inactivations often result in detrimental effects, due to its consequences for all cells. In this context, embryonic lethality represents the most severe detrimental effect when a gene essential for development is mutated. Consequently, the analysis of gene function at later stages is impeded or impossible. To solve this issue, conditional gene inactivation strategies have been developed. In organisms with efficient forward genetics, temperature-sensitive alleles have been generated[12,13]. Temperature-sensitive mutations are typically missense mutations, which retain protein function at permissive temperatures but fail to work properly at restrictive temperatures. In mammals, the most commonly used technique to achieve conditional gene inactivation employs the Cre/loxP system[14]. Cre recombinase promotes strand exchanges between two loxP target sites and depending on their orientation, recombination results either in the excision or inversion of the intervening DNA sequence. Thus, conditional gene inactivation can be achieved in a Cre-dependent manner if a gene or critical exon is flanked by loxP sites (floxed). Ligand-inducible Cre variants (CreER[T2]) offer additional temporal control of Cre-mediated recombination and allow targeting later aspects of a dynamic Cre expression[15]. CRISPR/Cas9 technology now also allows targeting loci with two loxP sites in other species like zebrafish[16,17]. However, establishment of a floxed allele and the generation of animals carrying the desired genetic composition is time and labor consuming. Moreover, although dual fluorescent gene labeling has been reported recently[18], floxed alleles are usually unlabeled impeding an easy recognition of mutant cells. Finally, Cre-mediated gene inactivation of floxed alleles requires two independent recombination events which is difficult to achieve in tissues with low recombination efficiencies[19]. Hence, we devised an alternative approach developing Cre-Controlled CRISPR (3C) mutagenesis as an easy and straightforward system that allows conditional gene inactivation in a Cre-dependent manner. 3C mutagenesis relies on a Cre effector construct with a promoter driving a floxed first open reading frame like a Stop cassette upstream of a second open reading frame encoding a Cas9-GFP fusion protein (Fig. 1a). The same transgenic construct expresses a gRNA targeting a gene of interest (GOI) under the zebrafish *U6a* promoter. In the default, unrecombined condition only the gRNA but no Cas9-GFP is present. Consequently, no functional Cas9/gRNA ribonucleoprotein complex is formed and the GOI remains intact. In contrast, an active Cas9/gRNA ribonucleoprotein complex is present in the cells after a successful Cre-mediated recombination event resulting in mutagenesis of the GOI.

Here, we show the functionality of this approach with stable transgenic lines, after applying it transiently to conditionally inactivate *cdh2* in the anterior neural plate[20]. As a proof-of-principle, we used a well-established target site in *tyrosinase*, the gene encoding the enzyme required for converting tyrosine into the pigment melanin. We demonstrate the functionality of 3C mutagenesis using Cre mRNA injections and various Cre/CreER[T2] driver lines, resulting in pigmentation loss in a Cre-dependent manner. Single-end NGS confirmed high-level mutagenesis in recombined cells. Taken together, our 3C conditional gene inactivation system is simple, fast, and allows gene inactivation in a Cre-dependent manner. Moreover, 3C mutagenesis might be scalable which would enable the conditional inactivation

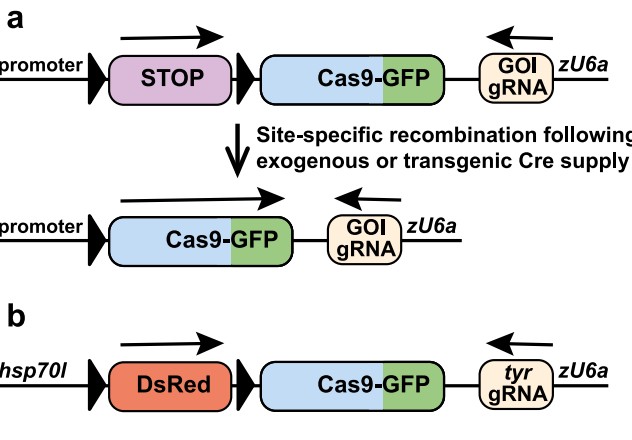

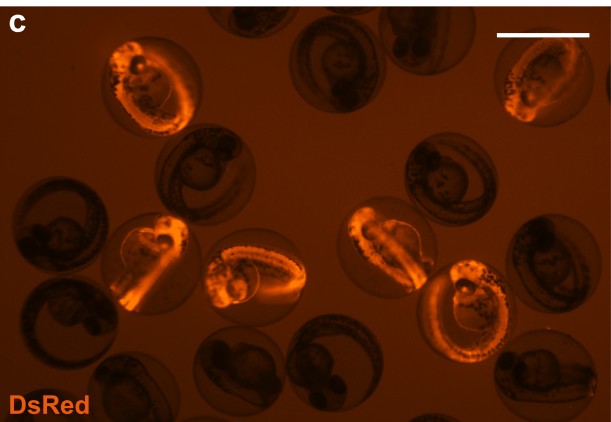

**Fig. 1 Cre-Controlled CRISPR (3C) mutagenesis allows gene inactivation in Cre-dependent manner. a** Scheme of the 3C rationale. A Cre effector construct controls the expression of a floxed Stop cassette upstream of the sequence encoding a fusion protein of Cas9 and GFP. In addition, a *U6a* promoter drives the constitutive expression of a gRNA targeting a gene of interest (GOI). Following exogenous or transgenic Cre supply, site-specific recombination results in the expression of Cas9-GFP. Combined with the gRNA a functional CRISPR complex is formed and mutates the target site within the gene of interest. **b** Scheme of the 3C gene inactivation construct targeting *tyrosinase* (*tyr*). The temperature-inducible *hsp70l* promotor drives expression of a floxed DsRed cassette. **c** Identification of transgenic animals expressing DsRed at 50 hpf after a heat treatment at 24 hpf. Example shown is a representative of a total of >100 heat-treated clutches from four independent 3C *tyr* transgenic insertions. Scale bar: 1000 μm.

of multiple genes simultaneously. Finally, following Cre-mediated recombination and expression of Cas9-GFP, presumptive mutant cells become fluorescently visible which enables the isolation of these cells and their subjection to various downstream omics techniques, like transcriptomics. Hence, 3C mutagenesis provides a valuable alternative to the production of floxed alleles and has the potential for applications in other model organisms amenable to single integration transgenesis.

## Results

**Establishment of a 3C gene inactivating line targeting tyrosinase.** In order to test the rationale of 3C, we generated a Tol2 transposon-based vector comprising the temperature-inducible *heat shock cognate 70-kd protein, like (hsp70l)* promoter controlling expression of a floxed DsRed cassette upstream of the coding sequence of a Cas9-GFP fusion protein (Fig. 1b). In addition, the vector contained a zebrafish *U6 promoter (U6a)* to drive transcription of a gRNA targeting *tyrosinase (tyr)*[8,21]. We chose to target *tyr* because it encodes the enzyme converting tyrosine into the pigment melanin. In zebrafish, *tyr* is expressed in cells of the retinal pigment epithelium (RPE) and in neural crest-derived melanocytes from 16.5 and 18 h postfertilization (hpf), respectively[22]. Consequently, mutagenesis of *tyr* prior to gene onset results in pigmentation defects in the developing eye and body, offering an easy readout of biallelic gene inactivation[8]. Using Tol2 transposon-mediated transgenesis[23], we identified 11 founders out of 21 animals screened using DsRed as a transgenesis marker present at 48 hpf after a 30 min heat treatment at 24 hpf (Fig. 1c). Four founders were used to establish stable transgenic lines (referred as 3C *tyr* henceforward).

**Cre mRNA injection results in ubiquitous GFP expression and widespread pigmentation loss.** Adult 3C *tyr* animals were crossed to wild-type and the resulting progeny were injected with in vitro transcribed Cre mRNA at the 1-cell stage, eliciting recombination during early stages of development (Fig. 2a). Following a heat treatment at 12 hpf, which activates Cas9-GFP-mediated mutagenesis in recombined cells, animals were analyzed at 22 and 50 hpf to score GFP expression and pigmentation, respectively. In comparison to uninjected controls, we observed a broad, ubiquitous GFP fluorescence in Cre mRNA injected animals at 22 hpf (Fig. 2b). At 50 hpf, GFP-positive embryos displayed a strong pigmentation loss within the developing eye and body highly similar to constitutive *tyr* mutants (Fig. 2c, d). Importantly, all animals showing GFP expression also displayed pigmentation defects if the heat treatment was applied prior to onset of *tyr* expression. For quantification, we determined the degree of pigmentation in the RPE using the gray value of black and white images of embryos expressing no fluorescent protein (No FP), non-recombined DsRed-positive (DsRed+), and recombined GFP-positive (GFP+) siblings as well as of constitutive *tyr* mutants (Supplementary Fig. 1 and Supplementary Table 1). No statistical difference was found when non-transgenic (No FP) and DsRed-positive 3C *tyr* siblings were compared. In contrast, GFP-positive 3C *tyr* animals were significantly different, similar to constitutive *tyr* mutants (Fig. 2e). To determine the mutation rate, we repeated the experiment to obtain GFP-positive cells from 30 Cre mRNA injected embryos and control cells from 30 siblings, respectively (Supplementary Fig. 2a). In order to ensure the analysis of the *tyr* locus from cells only with successful Cas9-GFP expression, we employed fluorescence-activated cell sorting (FACS) at 24 hpf which showed the presence of GFP-negative cells also in Cre mRNA injected embryos (Supplementary Fig. 2b). Following genomic DNA extraction from the FAC-sorted cells, the DNA was used as a template for a locus-specific

PCR amplifying the region targeted by the *tyr* gRNA. Subsequently, the amplified PCR fragments were used in single-end next generation sequencing (NGS) and the genome editing was analyzed using CRISPResso2[24]. In control cells, we found a single-nucleotide polymorphism in the first position of the protospacer adjacent motif (PAM). However, because any nucleotide is accepted in the canonical Cas9 PAM sequence NGG, this finding has no negative consequences with respect to targeting efficacy and only results in the presence of two parental sequences (Fig. 3a). In control cells, the parental strands can be detected with a proportion of 48.26% and 38.46%, respectively (Fig. 3b). In sharp contrast, the proportion of parental strands drops to 7.15 and 0.48% in GFP-positive cells and variable indels are abundantly present (Fig. 3c and Supplementary Fig. 3). Further global frameshift analysis revealed that 76.5% of the introduced indels represent frameshift mutations with deletions and insertions of up to 17 and 13 nucleotides, respectively (Fig. 3c′ and Supplementary Fig. 3b). The remaining 23.5% indels are in-frame mutations causing the deletion or insertion of a single or several amino acids which do most probably not interfere with protein function. We also analyzed off-target effects in the control and GFP-positive cells. The web tool for CRISPR- and TALEN-based genome editing CHOPCHOP[25] identifies one potential off-target on chromosome 19 with three mismatches to our employed *tyr* gRNA (Supplementary Fig. 4a). However, locus-specific amplification, single-end sequencing and CRISPResso2 analysis reveals highly similar allele frequencies with 83.1% in control and 88.55% in GFP-positive cells (Supplementary Fig. 4b). Taken together, these data show that our 3C *tyr* gene inactivation line robustly mediates biallelic gene disruptions in the RPE and in neural crest-derived melanocytes with no further side effects.

**Spatially controlled Cre activity results in tissue-specific GFP expression and restricted pigmentation loss.** To test gene inactivation in a tissue-specific manner, we combined our 3C *tyr* lines with Cre driver lines which allowed us to interfere independently with *tyr* activity in either RPE cells or in neural crest-derived melanocytes. To achieve the former, we made use of a knock-in of the inducible variant of Cre (CreER[T2]) into the endogenous *otx2b* locus[26]. For the latter, we used a transgenic line expressing Cre under a fragment of the zebrafish *sox10* promoter[27]. In *sox10*:Cre, Cre expression is initiated at 10 hpf resulting in Cre-mediated recombination in cells of the developing neural crest (Fig. 4a). Following a heat treatment at 16 hpf, we observed GFP fluorescence in neural crest cells at 22 hpf (Fig. 4b). At 50 hpf, neural crest-derived body pigmentation is present in control animals with no striking difference in GFP-positive embryos (Fig. 4c). However, closer examination revealed a reduction of pigmentation in the head region (Fig. 4d). This finding was further corroborated when GFP-positive animals were raised to adulthood. At this stage, mutant cells cover larger surface areas and displayed a loss of pigmentation in two vertical stripes as well as in the pectoral fins (Supplementary Fig. 5a). Because *sox10* expression is not initiated in all neural crest cells simultaneously, but in an anterior to posterior wave (Supplementary Fig. 5b), we hypothesized that the selected 6 h time window, in which Cre-mediated recombination and heat treatment-induced mutagenesis are happening, is too tight and hence causative for the mild observed pigmentation phenotype. To test this hypothesis, we repeated the experiment but provided three consecutive heat treatments at 16, 22, and 28 hpf to elicit gene inactivation in more cells (Supplementary Fig. 5c). Indeed, a further reduction in body pigmentation was observed (Supplementary Fig. 5d). Importantly, pigmentation in the RPE cells of the developing eyes was never affected in GFP-positive embryos neither in single nor

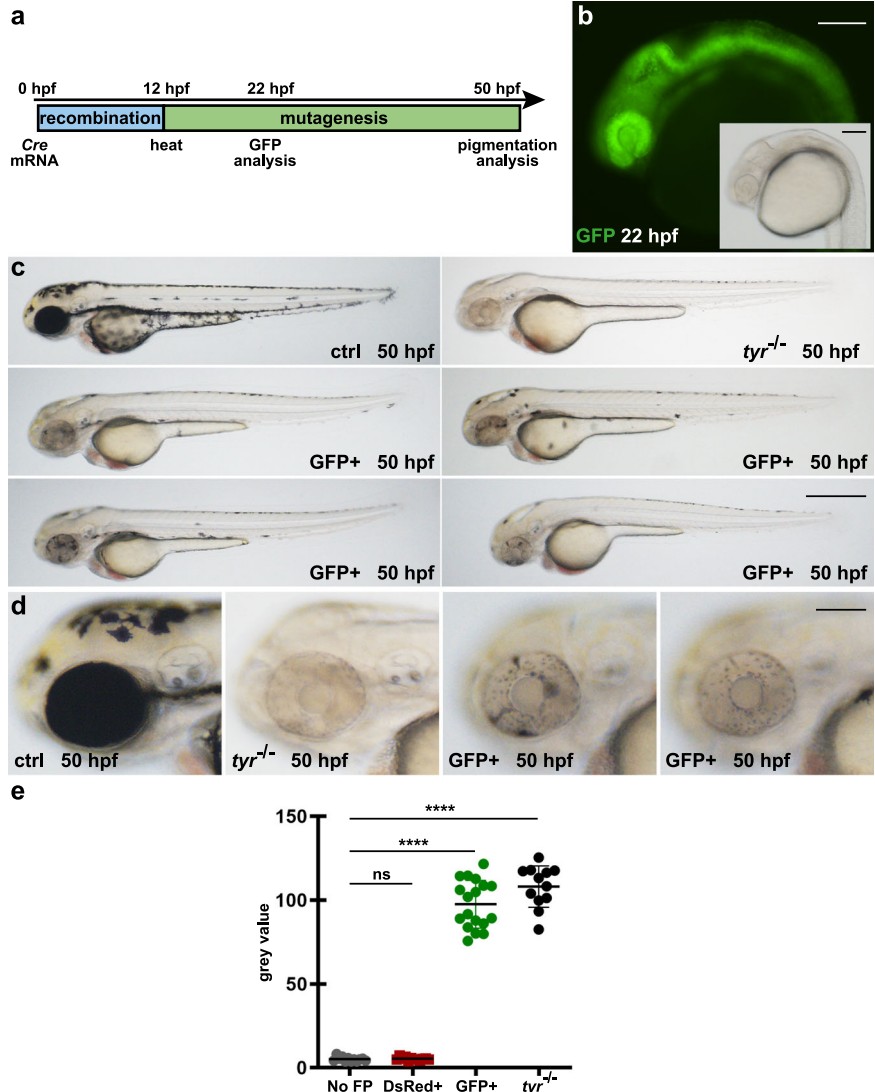

**Fig. 2 Cre mRNA injections into 3C *tyr* result in ubiquitous GFP expression and widespread pigmentation defects. a** Timeline. Cre mRNA injections at the 1-cell stage elicit ubiquitous recombination (blue box). At 12 hpf, a heat treatment triggers transient Cas9-GFP expression causing the subsequent permanent mutagenesis of the *tyr* target site (green box). Analysis of GFP expression and pigmentation was conducted at 22 and 50 hpf, respectively. **b** Expression of GFP is detected in a ubiquitous fashion at 22 hpf. In comparison to controls (ctrl), GFP-positive embryos (GFP+) display a significant loss of black pigment along the body (**c**) and in the developing eye (**d**). Examples shown are representatives across five experiments showing the same result. A total of >150 GFP-positive individuals and their respective non-GFP-positive siblings were analyzed. Scale bars: 150 µm in **b** and **d**; 500 µm in **c**. **e** Graph showing quantification of RPE pigmentation determined using the mean of the histogram from images of control embryos expressing no fluorescent protein (Non-FP), DsRed-positive (DsRed+), and GFP-positive (GFP+) siblings as well as *tyrosinase* mutants ($tyr^{-/-}$) at 50 hpf. $n$ (Non-FP, DsRed+, $tyr^{-/-}$) = 12; $n$ (GFP+) = 18. Data were analyzed using one-way ANOVA, followed by Tukey's post hoc test. Values are presented as mean ± SEM; ****$p \leq 0.0001$, ns (non-significant) >0.9999.

triple heat-treated animals. To achieve tissue-specific gene inactivation in RPE cells, we used *otx2b*:CreER[T2] in which a knock-in of the CreER[T2] coding sequence recapitulates the endogenous *otx2b* expression[26]. *otx2b* is initiated at 6 hpf and expressed in cells of the anterior neural plate including the eye field throughout early development. In order to induce widespread recombination, we applied 4-Hydroxytamoxifen (4OHT) at the onset of *otx2b* expression (Fig. 5a). Following a heat treatment at 12 hpf, strong GFP expression is observed in the anterior brain including the developing eyes, fore- and midbrain at 22 hpf (Fig. 5b). At 50 hpf, body pigmentation is indistinguishable in control and GFP-positive embryos (Fig. 5c). However, we observe a strikingly significant reduction of pigmented cells in the developing eye in GFP-positive embryos in comparison to controls (Fig. 5d). Importantly, GFP fluorescence and pigmentation defects were

only displayed after application of both, 4OHT and a heat treatment, but never elicited in the presence of 4OHT or heat treatment only. Quantitative comparison of non-transgenic with DsRed-positive animals revealed no significant difference in pigmentation of the RPE. In contrast, GFP-positive 3C *tyr* animals were significantly different and highly similar to constitutive *tyr* mutants (Fig. 5e and Supplementary Table 2). In summary, these data show that our 3C *tyr* lines can be used for tissue-specific biallelic gene inactivation in a conditional Cre-dependent manner.

**A recombined 3C *tyr* transgene allows temporally controlled gene inactivation and is transmitted to the next generation.** The above-mentioned experiments show that 3C gene inactivation can be used simply as a regular Cre effector line which, in

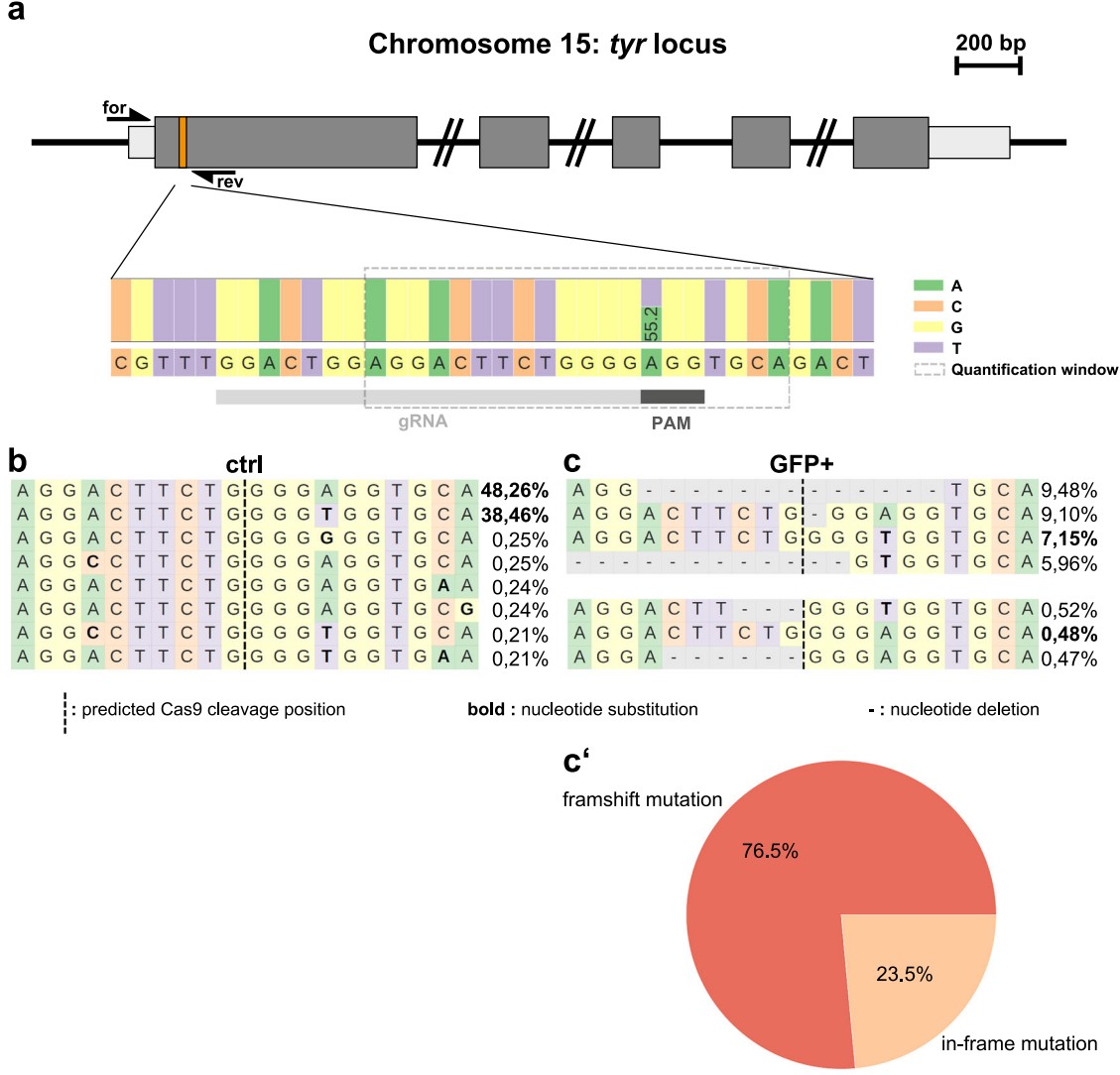

**Fig. 3 Sequencing confirms high mutagenesis rates following Cas9-GFP expression. a** Scheme of the *tyr* locus at chromosome 15. Exon sequences with translated and untranslated regions are represented in dark and light gray, respectively. Position of the CRISPR/Cas9 target in the first exon is indicated with an orange box. Forward and reverse primers used for amplification of the target site (for and rev) are shown as half arrows. Scale bar: 200 base pairs (bp). Next generation sequencing of FAC-sorted cells from pooled embryos (see Supplementary Fig. 1a) revealed a single-nucleotide polymorphism at the first position of the protospacer adjacent motif (PAM) indicated with a dark gray box downstream to the target site (gRNA) indicated by a light gray box. **b, c** Distribution of identified alleles around the cleavage site for the guide GGACTGGAGGACTTCTGGGG in control (ctrl) and GFP-positive cells (GFP+). Nucleotides are indicated by unique colors (A = green; C = red; G = yellow; T = purple). Substitutions are shown in bold font. Horizontal dashed lines indicate deleted sequences. The vertical dashed line indicates the predicted cleavage site. Sequencing of controls (**b**) shows presence of the parental strands with a frequency of 48.26 and 38.46% which drops to 7.15 and 0.48% in GFP-positive cells (**c**). **c'** Further sequence analysis of GFP-positive cells shows that 76.5 and 23.5% of the introduced indels represent frameshift and in-frame mutations, respectively.

combination with a Cre driver, allows spatiotemporally controlled mutagenesis of a GOI. However, because the number of available Cre driver lines is still limited in zebrafish, and moreover, experimental designs might require loss-of-function in the entire organism, we addressed ubiquitous, temporally controlled only gene inactivation. To do so, we made use of the conditional nature of the *hsp70l* promoter, which is only active after a heat treatment but not at permissive temperatures[28]. Hence, it should be possible to decouple recombination and mutagenesis if Cre activity but no heat treatment is applied. However, a basal leakiness has been reported for the *hsp70l* promoter and robust Cre-mediated recombination was observed with a *hsp70l:Cre* transgene even at permissive temperatures[29]. To test if *hsp70l* promoter provided basal levels of Cas9-GFP might cause non-conditional mutagenesis of the target gene, we repeated the Cre

mRNA injection experiment to elicit recombination at early stages (Fig. 6a). One subset of injected embryos was heat treated at 12 hpf, a second at 60 hpf and successful recombination was identified via GFP expression at 24 or 72 hpf, respectively. A third subset that never underwent a heat treatment was collected at 120 hpf. Because Cre mRNA injected 3C *tyr* transgenic animals are indistinguishable from wild-type siblings in the absence of a heat treatment, we established a 3C-specific PCR which even allowed the identification of recombined and non-recombined alleles (Supplementary Fig. 6a, b). This PCR strategy was applied to single embryos of all three time points (heat: none; heat: 12 hpf; heat 60 hpf) as well as various controls (wild-type, 3C *tyr*; no Cre, no heat treatment, 3C *tyr*; no Cre, with heat treatment) to verify the genotype of each embryo (Supplementary Fig. 6c). Subsequently, the genomic DNA of ten embryos per sample were

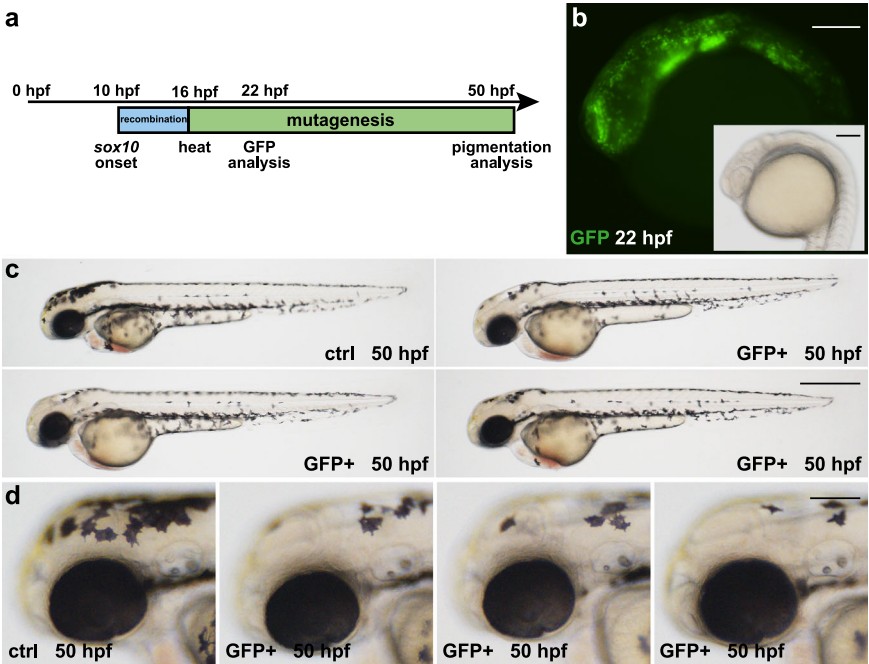

**Fig. 4 *sox10*-specific Cre activity results in spatiotemporally controlled GFP expression and pigmentation defects along the body. a** Timeline. Cre activity in *sox10*:Cre-positive animals at 10 hpf elicits recombination (blue box) in developing neural crest cells. At 16 hpf, a heat treatment triggers expression of Cas9-GFP and the subsequent permanent mutagenesis of the *tyr* target site (green box). Analysis of GFP expression and pigmentation was conducted at 22 and 50 hpf, respectively. **b** Expression of GFP is detected in neural crest cells at 22 hpf. **c, d** In comparison to controls (ctrl), GFP-positive embryos (GFP+) display a loss of pigmentation in the anterior head region. Examples shown are representatives across eight experiments showing the same result. A total of >160 GFP-positive individuals and their respective non-GFP-positive siblings were analyzed. Scale bars: 150 μm in **b** and **d**; 500 μm in **c**.

combined and analyzed using single-end NGS. Finally, GFP-positive siblings identified at 12 and 60 hpf as well as siblings never subjected to a heat treatment were raised to adulthood. Importantly, Cre mRNA injected animals, which were never subjected to a heat treatment, were indistinguishable from controls at embryonic and adult stages (Fig. 6b and Supplementary Fig. 7a, b). Also, sequencing of the *tyr* locus in the 120 hpf animals, which were exposed to Cre but no heat, confirmed an unmodified locus with a proportion of the parental strands at 76.80 and 14.10%. Genotyping of adult animals confirmed recombined 3C *tyr* alleles indicating its inactivity during growth. In sharp contrast, animals heat treated at 12 hpf, displayed a strong pigmentation phenotype at 50 hpf that persisted well into adulthood (Fig. 6c). Consistent with strong pigmentation defects observed at embryonic stages, the loss of pigmentation was also present in adult animals, although variable amounts of pigmented cells could be detected (Fig. 6c and Supplementary Fig. 7d). High-level mutagenesis was confirmed by NGS when variable indels were abundantly present and the proportion of the parental strands dropped to 20.64 and 0.85% (Fig. 6c and Supplementary Fig. 7e). Global frameshift analysis showed that 79.9% and 20.1% of the indels represented frameshift mutations and in-frame mutations, respectively (Supplementary Fig. 7e'). The proportion of parental strands in control samples (wild-type, 3C *tyr*; no Cre, no heat treatment, 3C *tyr*; no Cre, with heat treatment) was always >90% (Supplementary Fig. 7f). Animals heat treated at 60 hpf displayed naturally the normal pigmentation pattern at 50 hpf, identical to controls (Fig. 6d). However, variable pigmentation-loss was evident at adult stages (Fig. 6d and Supplementary Fig. 7b). Mutagenesis of the *tyr* target site was also confirmed via sequencing with the parental strands present with only 67.62 and 16.02%. To test if the recombined 3C *tyr* locus is successfully transmitted and can be reactivated again, we crossed animals exposed to Cre activity but no heat treatment with wild-type.

Subsequently, progeny underwent a heat treatment at 12 hpf and a strong and ubiquitous GFP expression was observed in 86% of the transgenic progeny at 22 hpf. Only 14% of the transgenic progeny displayed DsRed expression indicating the transmission of an unrecombined 3C *tyr* allele. As expected, we observe a strong reduction in pigmentation of the body and developing eye at 50 hpf (Fig. 6e). Taken together, these results show that the basal leakiness of the *hsp70l* promoter driving Cas9-GFP does not result in the production of enough gene product to elicit non-conditional mutagenesis. Hence, Cre-mediated recombination and Cas9-induced gene inactivation can be decoupled allowing temporally controlled mutagenesis of the target gene. Moreover, the recombined 3C *tyr* locus is stably inherited at a high frequency.

## Discussion

Previous reports showed that transgenic zebrafish with stable Cas9 and gRNA expression can be used to induce biallelic gene inactivation in somatic cells[21,30]. However, these approaches remained either non-conditional because Cas9 activity was only spatially controlled in a single transgene or genetically challenging because it required the presence of three different transgenes providing expression of a gRNA, a Cre-controlled Cas9 and Cre activity. Moreover, in neither of these cases Cas9 expressing and hence putative mutant cells were labeled. Here, we now introduce 3C mutagenesis to achieve conditional gene inactivation building on the binary Cre/loxP system. Binary systems offer the advantage that driver and effector lines are initially established independently and without any detrimental effects. In this context, a 3C gene inactivation line can be generated like any other Cre effector line which is easily achieved in many organisms, including zebrafish, via Tol2 transposon-mediated transgenesis[23]. Following transposon-mediated transgenesis, the 3C gene inactivation line requires Cre activity to delete the floxed first open

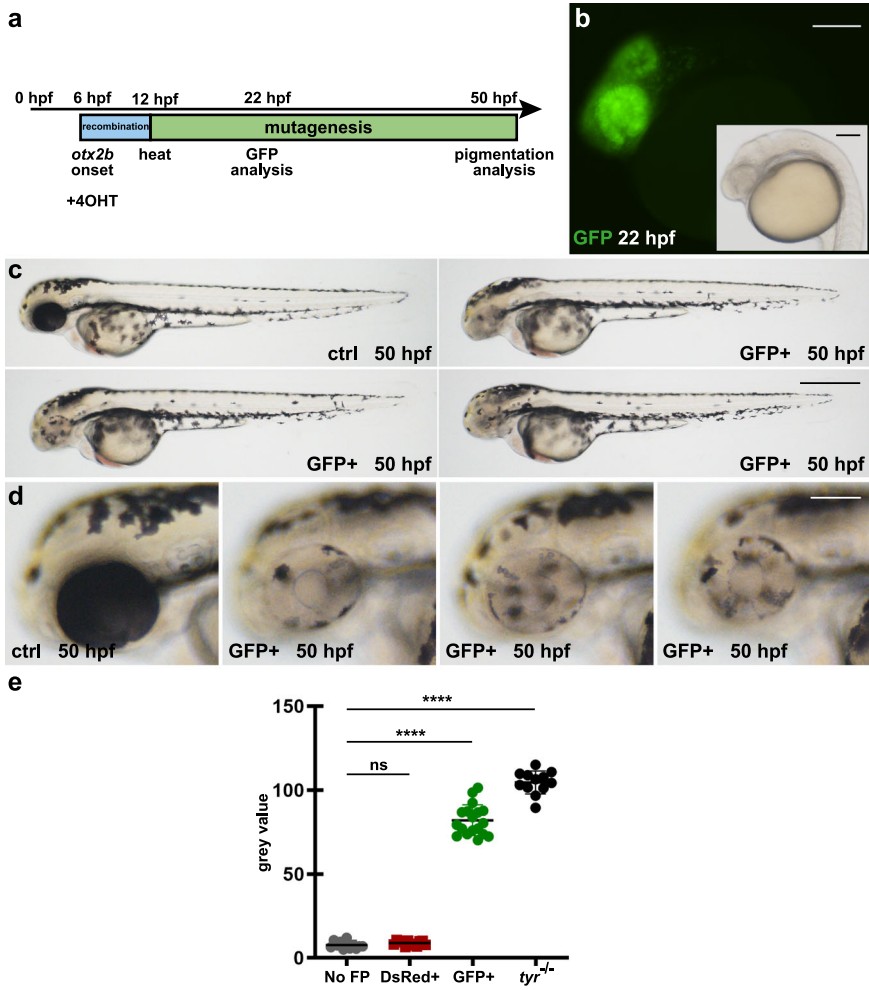

**Fig. 5 *otx2b*-specific Cre activity results in spatiotemporally controlled GFP expression and pigmentation defects in the developing eyes. a** Timeline. Application of 4-Hydroxytamoxifen (4OHT) induces Cre activity in *otx2b*:CreER^T2-positive animals at 6 hpf eliciting recombination (blue box) in cells of the developing anterior neural plate. At 12 hpf, a heat treatment triggers expression of Cas9-GFP and the subsequent mutagenesis of the *tyr* target site (green box). Analysis of GFP expression and pigmentation was conducted at 22 and 50 hpf, respectively. **b** Expression of GFP is detected in cells of the developing fore-, midbrain and eyes at 22 hpf. **c, d** In comparison to controls (ctrl), GFP-positive embryos (GFP+) display no pigmentation loss in cells along the body, but significant pigmentation defects in the developing eyes. Examples shown are representatives across ten experiments showing the same result. A total of >200 GFP-positive individuals and their respective non-GFP-positive siblings were analyzed. Scale bars: 150 μm in **b** and **d**; 500 μm in **c**. **e** Graph showing quantification of RPE pigmentation determined using the mean of the histogram from images of control embryos expressing no fluorescent protein (Non-FP), DsRed-positive (DsRed+), and GFP-positive (GFP+) siblings as well as *tyrosinase* mutants (*tyr*^−/−) at 50 hpf. n (Non-FP, DsRed+, *tyr*^−/−) = 12; *n* (GFP+) = 18. Data were analyzed using one-way ANOVA, followed by Tukey's post hoc test. Values are presented as mean ± SEM; ****$p \le 0.0001$, ns (non-significant) = >0.9999.

reading frame and to allow Cas9-GFP expression. In combination with the ubiquitously present gRNA, also driven from the same effector construct, a functional Cas9/gRNA ribonucleoprotein complex is formed triggering mutagenesis of the target site. In the best scenario, Cre activity is provided via a Cre driver line which allows precise spatial-temporal resolution in vivo as we showed with biallelic gene inactivation of *tyrosinase* in neural crest-derived melanocytes by using a *sox10*:Cre expressing transgene and in the developing eye by using an *otx2b*:CreER^T2 expressing transgene. However, also exogenously delivered Cre activity is possible. Potential approaches might include in vivo electroporation of Cre expressing plasmids or Cre transgene delivery via herpes simplex type I viruses into different organs at various developmental and adult stages[31,32]. We used Cre mRNA injections at the 1-cell stage to elicit recombination at the earliest time point possible which resulted in pigmentation defects in both neural crest-derived melanocytes and anterior neural plate-derived RPE cells. However, our phenotypic as well as sequence

analysis show that mutations can also be induced in a temporally controlled manner. At 50 hpf, Cre mRNA injected embryos displayed a normal pigmentation pattern and were indistinguishable from wild-type controls. Following a heat treatment at 60 hpf, indels were present in animals analyzed 12 h later and moreover, siblings raised to adulthood displayed various pigmentation defects. When compared to embryos heat treated at 12 hpf, the mutagenesis rate in embryos heat treated at 60 hpf was significantly reduced. However, this only indicates that the heat treatment applied at 12 hpf is not sufficient to induce indels in the same manner at 60 hpf. Hence, higher mutagenesis rates require adjustments to the heat treatment regime, which we have shown previously, can also be applied at adulthood[33]. Alternatively to Cre mRNA injections, a Cre driver can also be used for tissue-specific recombination at early stages followed by heat treatment to activate the *hsp70l* promoter at later stages to elicit mutagenesis. This way also some spatial control can be provided and mutagenesis is restricted to a certain lineage. Cre mRNA

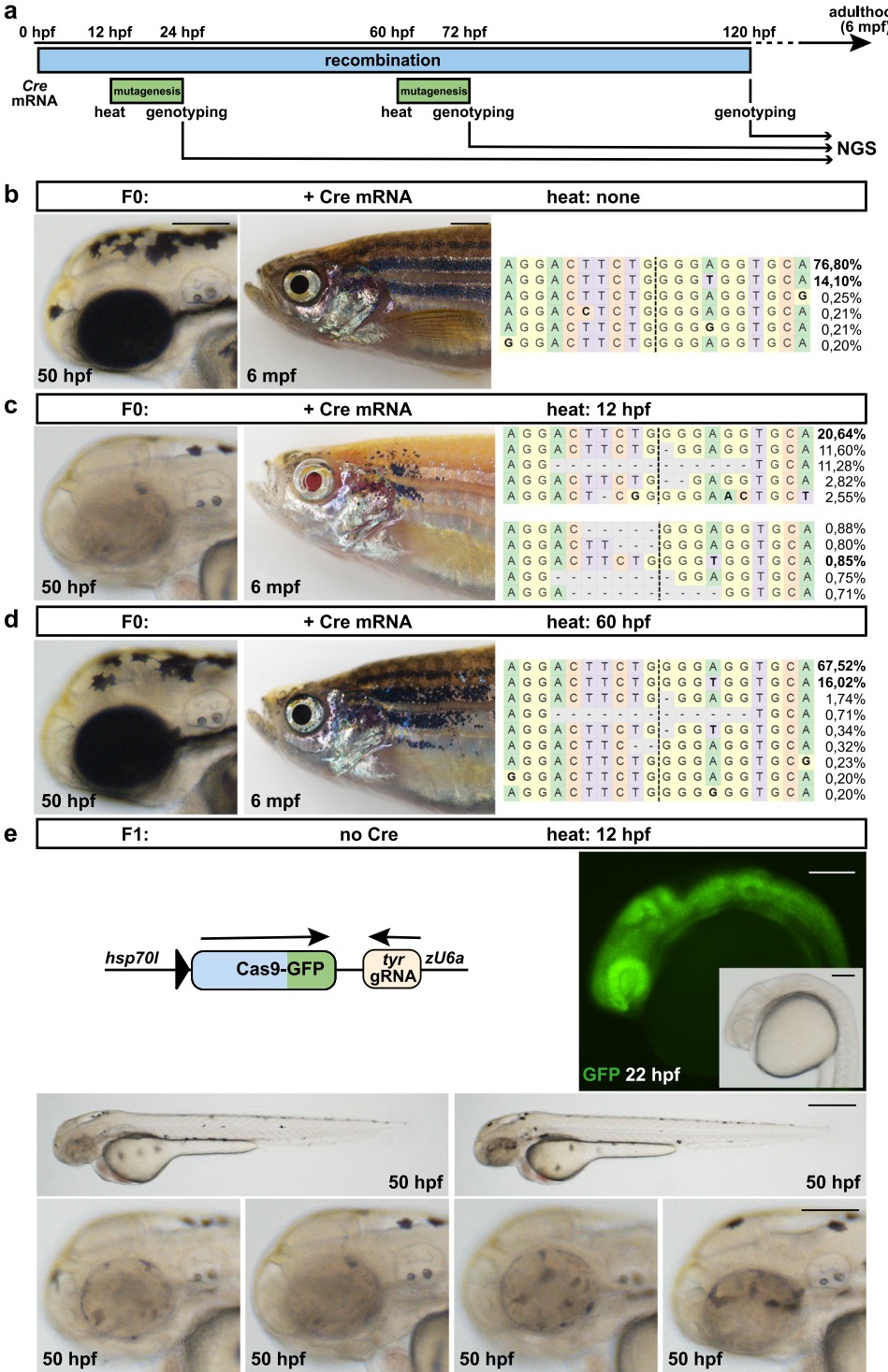

injections at the 1-cell stage can also be used to generate F1 animals carrying the recombined 3C gene inactivation construct in all cells. In our experiments, FACS revealed the presence of GFP-negative cells also in Cre mRNA injected embryos indicating that not all cells undergo recombination upon Cre mRNA delivery. Consequently, mutagenesis of the *tyr* target site cannot occur in non-recombined cells. This notion is supported by our sequencing data which show that the proportion of the parental strands in GFP-positive cells enriched via FACS increases from 7.63% (7.15 + 0.48%, Fig. 1) to 21.49% (20.64 + 0.85%, Fig. 6) in cells obtained from GFP-positive embryos without FACS enrichment. However, because recombined cells also contribute

to the germline, the recombined 3C gene inactivation construct is transmitted to the next generation. Here, it can be activated in a temporally controlled manner and, due to its presence in all cells, should theoretically result in a higher proportion of mutated target sites. In our proof-of-principle, we did not observe any obvious differences in pigmentation defects when comparing heat treated F0 animals subjected to Cre mRNA injection and heat-treated F1 animals carrying a recombined 3C *tyr* allele. However, because the selected *tyr* target site has been shown to be quite efficient[8], the potential increase might be hidden in the variable pigmentation phenotypes obtained. It is worth mentioning that F1 animals carrying a recombined 3C *tyr* allele are functionally

**Fig. 6 Temporally controlled gene inactivation. a** Timeline. Cre mRNA injections into progeny of 3C *tyr* animals at the 1-cell stage elicit ubiquitous recombination (blue box). A subset of injected embryos was heat treated at 12 or 60 hpf to trigger Cas9-GFP expression and mutagenesis of the *tyr* target site (green box). In addition, Cre mRNA injected animals were collected at 120 hpf without a prior heat treatment. Single embryo genotyping was applied to all specimen and ten embryos of each time point (heat: none; heat: 12 hpf; heat 60 hpf) were combined and analyzed using next generation sequencing (NGS). In addition, siblings of all three time points were raised to adulthood (6 months postfertilization (mpf)). **b** Animals subjected to Cre mRNA injection but no heat treatment show neither a pigmentation phenotype at embryonic or adult stages nor any signs of mutagenesis. **c** A strong pigmentation phenotype is observed at embryonic (50 hpf) and adult stages (6 mpf) in animals heat treated at 12 hpf. (Note almost complete absence of pigmentation in the RPE at 6 mpf making the eye appear red.) NGS confirms high-level indel production. **d** Pigmentation defects are observed at adult but not at embryonic stages in animals heat treated at 60 hpf. Sequencing corroborates mutagenesis of the *tyr* target site. Larval examples shown are representatives across three experiments showing the same result. Adult examples shown are representative of >15 individual fish examined. **e** The recombined 3C *tyr* allele is transmitted in the germline of F0 animals subjected to Cre mRNA injection only. Heat treatment in the F1 generation results in GFP expression and pigmentation defects after activation at 12 hpf. Examples shown are representatives across ten experiments showing the same result. A total of >300 GFP-positive individuals and their respective non-GFP-positive siblings were analyzed. Scale bars: 500 μm for 50 hpf and 2 mm for 6 mpf in **b**–**d**; 150 μm in **e**, except for lower row with 500 μm.

identical to fish in a recently published system allowing ubiquitous conditional gene ablation in zebrafish[34]. All aforementioned transgenic approaches employ a F0 CRISPR/Cas9-mediated mutagenesis strategy in which random indels are induced in parallel in different cells of the same organism. Several recent studies proved the efficiency of this strategy, in particular when more than one gRNA was used to target the GOI[35–37]. It is conceivable that the phenomenon of genetic compensation is not or only slightly induced in this strategy. Genetic compensation is triggered by mutant mRNA degradation and results in the upregulation of another gene or genes compensating the loss of an essential gene[38]. However, whereas a reduced or absent phenotype has been reported in several constitutive or germline knock-outs, the respective conditional gene inactivations displayed a more severe phenotype (reviewed in[39]). Further investigations will be necessary to address the role of genetic compensation in F0 CRISPR/Cas9-mediated mutagenesis approaches, including 3C.

Compared to the production of floxed alleles, our 3C gene inactivation offers several advantages. First, 3C mutagenesis is fast because preliminary loss-of-function analysis can already take place in the F1 generation when the 3C founder (F0) possesses a good transmission rate and is subsequently crossed to a Cre driver to obtain progeny carrying both transgenes. In contrast, time-consuming locus-specific genetic engineering is required for the generation of a floxed allele. Introduction of loxP sites flanking the entire locus or a critical exon is executed either sequentially or simultaneously in zebrafish[16,17]. However, even if the floxed locus is achieved with one round of genetic engineering, the subsequent breeding to obtain animals carrying the floxed locus in homozygosity and a Cre driver allele takes at least two generations. Second, putative mutant cells are genetically labeled due to the expression of Cas9-GFP. This will allow the implementation of lineage tracing studies of putative mutant cells in an otherwise wild-type context. Moreover, fluorescently labeled and putative mutant cells can be easily isolated using FACS. Their subsequent subjection to various omics techniques (e.g., transcriptomics) will provide additional details of cellular events in the absence of the GOI. In contrast, the mutant or wild-type status of floxed alleles is usually not provided via a fluorescent readout impeding an easy recognition of mutant cells. Only recently, a dual fluorescent gene labeling strategy has been reported, which however, still requires locus-specific genetic engineering[18]. Third, 3C gene inactivation should be scalable, which will enable the conditional inactivation of multiple genes simultaneously. In our proof-of-principle study, we used a single gRNA driven by the zebrafish *U6a* promoter to target the open reading frame in the first exon of *tyrosinase*. A single gRNA was used because the target site has already been shown to be quite efficient[8]. For future 3C gene inactivation lines,

it is recommended to employ two gRNAs because two guides are more effective in disrupting gene function than a single gRNA due to induction of small genomic deletions, in addition to frameshifts caused by indel mutations[36]. Expression of a second, third, and fourth gRNA is achieved using additional *U6* promoters that have been analyzed for this purpose[21] and that can be easily added into our 3C gene inactivation construct. Because the targeting efficiency of the guides is the most critical aspect in 3C gene inactivation, several guides should be tested transiently as recently described[35]. Subsequently, the two most efficient guides should be selected for the generation of the 3C gene inactivation construct employed in Tol2 transposon-mediated transgenesis. In contrast to potential inactivation of multiple genes via 3C, conditional gene inactivation with two or even more floxed alleles is theoretically feasible. However, generation of the floxed loci and subsequent breeding to obtain the desired genetic composition is very time consuming. Moreover, unintentional inter-chromosomal recombination can cause unwanted, detrimental side effects. Finally, a single recombination event allows for the expression of Cas9-GFP in our 3C gene inactivation system which is beneficial in tissues with low recombination efficiencies. In this context, using Cre/loxP-based genetic lineage tracing, we previously showed that *her4.3*-positive ventricular radial glia cells react to injury and generate new neurons in the zebrafish telencephalon[33]. Despite broad expression of CreER^T2 in a large proportion of the ventricular zone, we observed only limited recombination, most likely due to low local concentrations of 4OHT. Although, low recombination will persist also with 3C, the number of putative mutant cells will be identical to the number of cells observed in the genetic lineage tracing experiment and we will be able to follow their fate in a wild-type context due to Cas9-GFP expression. In contrast, Cre-mediated gene inactivation of floxed alleles requires two independent recombination events to elicit the mutant situation. Consequently, the number of mutant cells would be significantly further reduced in our tissue of interest. In conjunction with the absence of a genetic label, it will be almost impracticable to recognize mutant cells. Taken together, 3C mutagenesis will allow the fast setup of conditional gene inactivation experiments with a high penetrance which depends on (1) Cre-mediated site-specific recombination to allow Cas9-GFP expression and (2) the subsequent production of frameshift mutations. Whereas the former is an invariable property of the Cre supply, the latter can be improved by the use of two or even more gRNAs because the proportion of frameshift mutations increases with every additional gRNA as recently shown[36].

Our 3C will allow conditional gene inactivation studies and represents a valuable alternative to the production of floxed alleles requiring only the generation of one transgenic line. However, Tol2 transposon-mediated transgenesis usually results in the

integration of multiple copies of transgenes at random loci within the genome[23]. Subsequently, time-consuming characterization is required to identify integrations displaying the desired properties like expression level and pattern. Targeted, site-directed transgene integration into predetermined genomic loci can circumvent these issues. The integrase PhiC31 catalyzes an unidirectional recombination between heterotypic *attP* and *attB* sites and its functionality has already been shown in zebrafish[40]. In the long run, it will be beneficial to establish an *attP* landing site line in a safe harbor locus for 3C-mediated gene inactivation. 3C is a composite of a generic Cre effector construct that switches to Cas9-GFP after a successful recombination event and a U6 promoter cassette expressing a gRNA for gene-specific mutagenesis. Integration of the *attP* sequence into the generic 3C effector construct will result in a general platform for future 3C gene inactivation lines. The 3C landing site line needs to undergo thorough characterization only during its establishment. Subsequently, the *attP/attB*-mediated integrations can be directly used in a planned experimental setup because no further negative positional effects are expected. We foresee that the generation of a 3C landing site line will be highly useful. In addition to general benefits of landing site lines like reduction in time, cost, and experimental animals, the assembly of the *attB*-containing U6 promoter cassette is significantly simplified compared to the assembly of the entire construct containing the generic portion as well as the gene-specific U6 promoter cassettes. Moreover, the 3C landing site line would be compatible with other transgenic approaches for the expression of gRNAs. Recently, a tRNA-based multiplex gRNA expression system was shown to express up to ten different gRNAs from one scaffold[41]. Finally, the 3C landing site line would serve as the best possible control to a 3C landing site gene inactivation line because both lines utilize the same transgene integration site.

Another highly efficient system, termed CRISPR-Switch, also allows 3C/Cas9 mutagenesis[42]. However, in contrast to 3C mutagenesis, which enables the expression of Cas9 following a Cre recombination event, CRISPR-Switch controls the production of the gRNA in a Cre-dependent manner. In doing so, CRISPR-Switch allows even sequential gene editing of two loci in a temporal order, which is currently not possible with 3C mutagenesis. Despite this limitation, 3C mutagenesis has the potential for applications in other model organisms. In this context, we noted in the course of our studies that a similar system to 3C mutagenesis has been applied in mouse resulting in conditional gene inactivation[43]. However, only low Cas9 expression was reported and moreover, although an in-frame fusion with a FLAG tag allows subsequent identification of putative mutant cells via immunohistochemistry in fixed tissues, additional options like live imaging and FACS are not possible with the published design. We foresee that our 3C mutagenesis strategy will enable a variety of possible applications for conditional gene inactivation studies when used in combination with the appropriate Cre driver lines. In this context, recent efforts to increase the number of available Cre driver lines are now reinforced by new genome editing approaches allowing the targeted knock-in of Cre or CreER[T2] at endogenous loci in the zebrafish genome providing Cre driver lines[26,44]. These lines, which reliably recapitulate endogenous gene expression patterns, will help to exploit the full potential of 3C mutagenesis.

## Methods

**Ethical statement**. Fish were kept according to FELASA guidelines[45]. All animal experiments were conducted according to the guidelines and under supervision of the Regierungspräsidium Dresden (permit: TVV 21/2018). All efforts were made to minimize animal suffering and the number of animals used.

**Zebrafish husbandry and lines**. Zebrafish were kept and bred according to standard procedures[46,47]. Our experiments were carried out using zebrafish from wild-type stocks with the AB genetic background and the transgenic lines *Tg (otx2b:CreER^T2)tud44*, abbreviated *otx2b*: CreER^T2 [26] and *Tg(-4.7sox10:Cre)ba73* abbreviated *sox10*:Cre[27]. Combinations of the Cre driver lines with 3C *tyr* was conducted at least eight times with at least 20 double transgenic animals analyzed. All animals showing GFP expression also displayed pigmentation defects if the heat treatment was applied prior to onset of *tyr* expression. Constitutive *tyrosinase* mutants (*tyr^−/−*) were obtained via incrossing of adult 3C *tyr* animals subjected to Cre mRNA injection and heat treatment at 12 hpf (shown in Fig. 6c and Supplementary Fig. 7c).

**Plasmid construction and germline transformation**. To create the pTol hsp70l: loxP-DsRed-loxP-Cas9-GFP; U6a:tyr plasmid, several intermediate steps were taken. Initially, the coding sequence of GFP was PCR amplified from pTol hsp70l: loxP-DsRed-GFP[33] with primers GFP-for and GFP-rev (Supplementary Table 3) flanked by the unique restriction sites AgeI and XbaI, respectively. After digestion, the PCR product was cloned into the vector pT3TS nCas9n (addgene #46757)[8] replacing GFP and giving rise to pT3TS nCas9n-GFP. Subsequently, Cas9-GFP was digested using NcoI following blunt-ending with Klenow and a second digest with NheI. The fragment was subsequently ligated into pTol hsp70l:loxP-DsRed-GFP digested with SmaI and NheI replacing GFP and giving rise to pTol hsp70l:loxP-DsRed-loxP-Cas9-GFP (addgene #158962). To generate the U6a gRNA expression cassette, the *U6a* promotor including a gRNA backbone was PCR amplified from pDestTol2CG2-U6:gRNA (addgene #63156)[30] with primers U6a-for and U6a-rev (Supplementary Table 3) flanked by the unique restriction sites Acc65I and SacI, respectively. After digestion, the PCR product was cloned into the pBluescript giving rise to pBS U6a gRNA (addgene #158963). For the generation of pBS U6a *tyr*, the two BseRI enzyme sites at the 5′ end of the gRNA scaffold and the annealed oligos tyr-for and tyr-rev (Supplementary Table 3) covering the *tyrosinase* target site 5′-GGACTG-GAGGACTTCTGGGGAGG-3′[8] were used according to established protocols[30]. Finally, pBS U6a tyr was digested using AscI and NheI and ligated into pTol hsp70l:loxP-DsRed-loxP-Cas9-GFP giving rise to final pTol hsp70l:loxP-DsRed-loxP-Cas9-GFP; U6a:tyr plasmid which was subsequently used for germline transformation. To this aim, plasmid DNA and transposase mRNA were injected into fertilized eggs (F0), raised to adulthood, and crossed to AB wild-type fish as previously described[23]. To identify transgenic carriers, undechorionated F1 embryos were heat shocked at 24 hpf and examined under a fluorescent microscope at 50 hpf. This way, 11 out 21 independent founders were identified and four founders were chosen to establish independent lines (referred as 3C *tyr*). For detection of the recombined and non-recombined 3C construct, the forward primer hsp70l-for and the reverse primer Cas9-rev (Supplementary Table 3) were used in a standard PCR on genomic DNA (PCR parameters: 2 min at 95 °C followed by 35 cycles with 30 s at 95 °C, 30 s at 59 °C, and 30 s at 72 °C followed by 7 min at 72 °C). All newly generated plasmids are available from addgene.org. 3C *tyr* transgenic lines will be provided upon request.

**4-Hydroxytamoxifen, Cre mRNA, and heat treatments**. For 4OHT (Sigma, H7904) treatments a 25 mM stock solution was made and stored at −20 °C. Embryos, still in their chorions, were transferred into petri dishes containing 4OHT with a working concentration of 0.5 μM in early gastrulation (shield or 6 hpf). For control treatments, sibling embryos were incubated in corresponding dilutions of ethanol. All incubations were conducted in the dark. Cre mRNA was generated from *pCS2 + Cre* using linearization with NotI and the mMESSAGE mMACHINE SP6 kit (Invitrogen) according to the manufacturer's instructions[48]. Cre mRNA injections into 3C *tyr*-positive progeny at the 1-cell stage was conducted five times with at least 30 transgenic animals analyzed. All animals showing GFP expression also displayed pigmentation defects if the heat treatment was applied prior to onset of *tyr* expression. Importantly, animals heat treated later but still prior to first signs of pigmentation (22 hpf), also showed strong GFP expression but displayed a normal pigmentation pattern. For heat treatments, embryos, still in their chorions, were transferred into fresh petri dishes. After removal of excess embryo medium, 42 °C embryo medium was added and the petri dishes were kept for 30 min in a 37 °C incubator before they returned to a 28.5 °C incubator.

**Imaging**. Embryos and larvae were anesthetized with 0.01% MESAB (MS-222 or tricaine) in E3 medium and then mounted in 3% methylcellulose. Images were taken by a Olympus MVX microscope equipped with Olympus DP80 digital camera and the cellSens Dimension imaging software. Images were processed using Adobe Photoshop CC2015. Figures were assembled using Adobe Illustrator CC2015.

**Tissue dissociation and fluorescence-activated cell sorting (FACS)**. Tissue dissociation was conducted as described previously[49]. Briefly, embryos were removed from their chorions by pronase treatment[46], followed by deyolking at 4 °C in 0.5% Ginzburg-Ringer without CaCl₂. Dissociation was conducted in trypsin-EDTA on ice. When embryos were completely dissociated, the reaction was

stopped by adding Hi-FBS. The cells were pelleted, washed with PBS, resuspended in PBS, and passed through a 40 μM mesh filter prior to cell sorting. FACS was performed using an Aria II cell sorter (BD Biosciences). Forward and side scatter were used to gate for live, single cells, out of which GFP-positive cells were sorted and collected. For control cells from GFP-negative embryos the same gating strategy was employed. Flow cytometry data were analyzed using BD FACSDiva software. DNA was extracted from sorted cells using Quick-gDNA Miniprep Kit (Zymo Research, Cat. No.: D3025) following manufacturer's instructions for cell suspensions.

**High-throughput single-end sequencing and CRISPR/Cas9 genotyping.** Genomic DNA from individual embryos/larvae was extracted according to the "still quick, less dirty" protocol[46]. The regions containing the *tyr* target or *tyr* off-target were amplified in a standard PCR (PCR parameters: 2 min at 95 °C followed by 25 cycles with 30 s at 95 °C, 30 s at 59 °C, and 30 s at 72 °C followed by 7 min at 72 °C) using tyr-P5-tail and tyr-P7-tail or tyr-off-P5-tail and tyr-off-P7-tail (Supplementary Table 3), respectively. After amplification, PCR products were purified using NuceloSpin Gel and PCR Clean-up (Macherey Nagel, 740609.250) following the manufacturer's instructions. In order to sequence the CRISPR PCR products, an 8-cycle index PCR with Illumina TruSeq-primer was run with 2 ng starting material. Libraries were quantified, and 75 bp single-end reads were sequenced on the Illumina NextSeq 500 platform to a minimum depth of 4 million reads. We used the tool CRISPResso2[24] to trim sequencing adapters, align sequencing reads to the amplified PCR sequence, quantify the number of insertions, substitutions and deletions in the reads, and finally visualize the results. Default parameters were applied with the exception of *--trim_sequences* and *--trimmomatic_options_string* to trim sequencing adapters, *--quantification_window_size 10*, and *--exclude_bp_from_left 0 --exclude_bp_from_right 0*.

**Quantification of RPE pigmentation and statistical analysis.** To determine the degree of pigmentation in the RPE, 8bit black and white images of embryos expressing No FP, DsRed- and GFP-positive siblings as well as *tyrosinase* mutants ($tyr^{-/-}$) were taken at 50 hpf. Following the selection of the region of interest, covering the dimension of the developing eye, the defined area was applied manually to all images and the mean of the histogram was measured using FIJI (ImageJ). For statistical analysis, one-way ANOVA, followed by Tukey's post hoc test was performed. Values are expressed as mean ± SEM. Graphs were performed using PRISM7 (Graph-Pad) software. Error bars represent standard error of the mean (SEM); ****$p \le 0.0001$, ns (non-significant) = >0.9999.

**Reporting summary.** Further information on research design is available in the Nature Research Reporting Summary linked to this article.

## Data availability

No datasets were generated or analyzed during the current study. Source data are provided with this paper (Supplementary Tables 1 and 2). All sequencing data that support the findings of this study have been deposited in the NCBI Sequence Read Archive and are accessible through BioProject ID PRJNA693993. All other relevant data are available from the authors upon reasonable request.

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

## Acknowledgements
We thank Drs. W.B. Chen and L. Zon for sharing reagents, M. Fischer, J. Michling, and D. Mögel for excellent zebrafish care and Drs. R. Behrendt, S. Schulte-Merker, and R. Kelsh for critically reading the manuscript. In addition, we thank the members of the Brand laboratory for continued support and discussions as well as helpful comments on the manuscript. This work was supported by the Light Microscopy Facility and Flow Cytometry Facility, core facilities of the CMCB at the Technische Universität Dresden. Funding was provided by the Technische Universität Dresden, the Deutsche Forschungsgemeinschaft to S.H. (HA 6362/2) and M.B. (BR 1746/3) and the European Union (European Research Council AdG Zf-BrainReg) to M.B. The funders had no role in study design, data collection and analysis, decision to publish, or preparation of the manuscript.

## Author contributions
Conceptualization: S.H.; Investigation: S.H., D.Z., J.H., J.S., S.S., V.K., J.S.E., D.E., A.P., and A.D; Resources: G.K.; Writing—original draft: S.H.; Writing—review and editing: J.H., G.K., V.K., J.S.E., D.E., and M.B; Supervision: S.H.; and Funding acquisition: S.H. and M.B.

## Funding

## Competing interests
The authors declare no competing interests.
