## [Peer Review File · Nature Communications]

Reviewers' Comments:

Reviewer #1:

Remarks to the Author:

This manuscript by the Hans and Brand groups describes a new method for cell type specific conditional gene ablation, they call 3C. Rather than introducing loxP sites flanking the gene of interest in the fish genome, the authors express, from one plasmid a floxed stop codon in front of a cas9-GFP fusion construct and a ubiquitous gRNA against a gene of interest. Introduction of cre by cre mRNA injection or mating fish with fish expressing cre or creERT2 under a cell type specific promoter leads to the excision of the stop codon, expression of cas9 and formation of an active CRISPR complex. Since the cells in which the cas9-GFP transgene is expressed are fluorescent, mutated cells can be isolated from the animals by FAC sorting allowing further analyses. The authors illustrate the different induction possibilities interfering with one gene of known function in fish of different ages. Furthermore, they sequence the mutations resulting from their manipulations showing effectiveness of their methods. Cell type specific mutation using a single plasmid had been described before in Ablain et al., 2015, but the conditional induction hadn't. This is a very useful addition to the methods available to manipulate gene expression in the fish and explore gene function specific to a cell type at a chosen developmental time, without risking to perturb the development of the fish due to early broader gene function.

The results are very thoroughly analysed and systematically presented. The discussion is maybe a bit idiosyncratic and the biological novelty is limited, since the paper only tests the function of one known gene.

Given the effectiveness of the method, could the cas9/gRNA construct be injected into cre lines to analyse gene function in the F0?

Abstract: In the abstract, it is unclear how the method works. The sentence starting "In contrast" (line 24ff) is illogical.

Line 54ff: be clearer that conditional mutagenesis is lethal when genes essential for early development are mutated, precluding analyses of gene function for later events.

Fig 1: the schematic should make clearer how cre can be introduced.

Reviewer #2:

Remarks to the Author:

This is a nice proof of principle study. The 3C conditional mutagenesis strategy is a clever idea – moving CRISPR mediated knockout into a system that implements multiple well-established strategies for transgene control, notably Cre recombination and heat-shock induction. The authors focus their characterization in zebrafish, which I believe this technology will be a great interest to this community, and the potential of this approach, with modification, can easily span models. The authors display the efficacy of their system using a KO proof staple, the tyr locus. They demonstrate loss of pigment using both global and promoter driven Cre recombination coupled to heat shock induction of the tagged Cas9. Using illustrative phenotype examples and next gen sequencing, the authors show phenotype and genotype disruption, respectively. A notable advantage of the 3C system is the presence of fluorescent indicators for pre and post recombined states. As the authors note and have applied as a validation tool, this feature aids genomic strategies for analysis in a rigorous manner. Overall, this is a very nice and well executed study. In order to clearly and rigorously conceptualize 3C efficacy and to clarify on a point that appears to go against established zebrafish dogma, I have several suggestions/comments that I believe would improve the manuscript and provide information highly relevant to labs interested in applying this approach.

Major:

1. Phenotype characterization is largely observational. In order to appreciate the range of phenotype penetrance, some categorical quantification is needed comparing density (or other parameter) of pigmentation in 3C ablated individuals, WT, and tyr mutants. This is not needed for all iterations yet quantification for the Cre mRNA injected 12 hpf HS or for the RPE KO would be useful and provide statistical rigor.
2. As this is a F0 mutagenic strategy generating random genotypes – quantifying the rate of potentially functional in-frame mutations from your sequencing is needed. Discussing this in combination with a gross estimate of non-recombined loci would be helpful to better estimate the penetrance of 3C (IE non-recombined due to variable Cre mRNA expression or mosaicism of Cre driving transgenes). Demonstration of 3C effectiveness with a different gDNA target, alternative gene or target within tyr itself, would be ideal to highlight penetrance beyond an N=1, yet arguably beyond the scope of the current study.
3. Can the authors address the lack of leakiness from the pHSP1 promoter? The basal promoter activity of pHSP1 in zebrafish is well established and this promoter is used as a minimal driver for enhancer traps screens. The authors should use a 3 primer test, similar to Sup 5C, with an additional primer in dsRed and show a low and high exposure gel, in order to absolutely confirm the complete absence of non-induced gene disruption, as the authors state. This is important to rigorously address 3C stability and therefore the reliability of control experiments. If the authors confirm that their construct is absolutely not leaky, they should comment as to the cause – insertion position, which would be a great interest as well.

Minor

1. The authors assert that 3C is scalable and viable for multi target knockout, yet this is not currently demonstrated. Therefore, in the abstract this should not be a definitive statement, yet I agree an apt point for discussion.
2. The authors should discuss their system in relationship to the heat shock induced CRISPR ablation recently published:
Wu YC, Wang IJ. Heat-shock-induced tyrosinase gene ablation with CRISPR in zebrafish. *Mol Genet Genomics*. 2020 Jul;295(4):911-922. doi: 10.1007/s00438-020-01681-x. Epub 2020 May 4. PMID: 32367255.
3. Line 210. Figure 4A may not be the correct figure reference.
4. Including pictures of tyr mutants in some figures would be helpful to highlight knockout phenotype
5. Is there a proposed reason the parent allele percentages are dramatically divergent (notable Sup 6F)?
6. Some discussion of studies that used a F0 CRISPR mediated mutagenic strategy(1–3) would be helpful and highlight the applicability of this approach.
7. Can the authors envision scenarios where 3C would not be ideal – highly restricted patterns of expression etc? Adding this to the discussion would be valuable insight to labs interested in applying 3C.
8. Line 345 – can a reference be provided for low recombination tissue?
9. Can the authors provide a rate for F1 transmission following Cre recombination?

10. The authors discuss using phi landing sites for 3C constructs. Out of curiosity, do the authors think using similar techniques flanking the gRNA locus specifically would be useful for rapid gene target swapping?

11. Addressing CRISPR knockout mediated compensation(4) in the discussion would be immensely helpful and an important issue to consider when applying 3C to different genetic targets.

1. Wu, R. S. et al. A Rapid Method for Directed Gene Knockout for Screening in G0 Zebrafish. *Developmental Cell* 46, 112-125.e4 (2018).

2. Shah, A. N., Davey, C. F., Whitebirch, A. C., Miller, A. C. & Moens, C. B. Rapid reverse genetic screening using CRISPR in zebrafish. *Nature Methods* 12, 535-540 (2015).

3. Horstick, E. J., Bayley, Y., Sinclair, J. L. & Burgess, H. A. Search strategy is regulated by somatostatin signaling and deep brain photoreceptors in zebrafish. *BMC Biol.* 15, 4 (2017).

4. El-Brolosy, M. A. et al. Genetic compensation triggered by mutant mRNA degradation. *Nature* 568, 193-197 (2019).

Reviewer #1 (Remarks to the Author):

This manuscript by the Hans and Brand groups describes a new method for cell type specific conditional gene ablation, they call 3C. Rather than introducing loxP sites flanking the gene of interest in the fish genome, the authors express, from one plasmid a floxed stop codon in front of a cas9-GFP fusion construct and a ubiquitous gRNA against a gene of interest. Introduction of cre by cre mRNA injection or mating fish with fish expressing cre or creERT2 under a cell type specific promoter leads to the excision of the stop codon, expression of cas9 and formation of an active CRISPR complex. Since the cells in which the cas9-GFP transgene is expressed are fluorescent, mutated cells can be isolated from the animals by FAC sorting allowing further analyses. The authors illustrate the different induction possibilities interfering with one gene of known function in fish of different ages. Furthermore, they sequence the mutations resulting from their manipulations showing effectiveness of their methods. Cell type specific mutation using a single plasmid had been described before in Ablain et al., 2015, but the conditional induction hadn't. This is a very useful addition to the methods available to manipulate gene expression in the fish and explore gene function specific to a cell type at a chosen developmental time, without risking to perturb the development of the fish due to early broader gene function.

The results are very thoroughly analysed and systematically presented. The discussion is maybe a bit idiosyncratic and the biological novelty is limited, since the paper only tests the function of one known gene.

Given the effectiveness of the method, could the cas9/gRNA construct be injected into cre lines to analyse gene function in the F0?

Yes, 3C mutagenesis can also be applied transiently. We actually used this approach and injected a 3C construct into progeny carrying an *otx2b:CreERT²* transgene to inactivate *cdh2* in the anterior neural plate conditionally (Kesavan et al., 2020). We rephrased this subject and now refer to this approach in the introduction (line 83ff).

Abstract: In the abstract, it is unclear how the method works. The sentence starting "In contrast" (line 24ff) is illogical.

The reviewer is right that we did not mention the methodology of 3C in the abstract. Because reviewer 2 raised an additional issue, we significantly revised large parts of the abstract (line 23ff).

Line 54ff: be clearer that conditional mutagenesis is lethal when genes essential for early development are mutated, precluding analyses of gene function for later events.

As the reviewer suggested, we rephrased and made the statement more clearly in the introduction (line 52ff).

Fig 1: the schematic should make clearer how cre can be introduced.

We followed the advice of the reviewer and adjusted the figure as well as its legend appropriately.

Reviewer #2 (Remarks to the Author):

This is a nice proof of principle study. The 3C conditional mutagenesis strategy is a clever idea – moving CRISPR mediated knockout into a system that implements multiple well-established strategies for transgene control, notably Cre recombination and heat-shock induction. The authors focus their characterization in zebrafish, which I believe this technology will be a great interest to this community, and the potential of this approach, with modification, can easily span models. The authors display the efficacy of their system using a KO proof staple, the *tyr* locus. They demonstrate loss of pigment using both global and promoter driven Cre recombination coupled to heat shock induction of the tagged Cas9. Using illustrative phenotype examples and next gen sequencing, the authors show phenotype and genotype disruption, respectively. A notable advantage of the 3C system is the presence of fluorescent indicators for pre and post recombined states. As the authors note and have applied as a validation tool, this feature aids genomic strategies for analysis in a rigorous manner. Overall, this is a very nice and well executed study. In order to clearly and rigorously conceptualize 3C efficacy and to clarify on a point that appears to go against established zebrafish dogma, I have several suggestions/comments that I believe would improve the manuscript and provide information highly relevant to labs interested in applying this approach.

Major:

1. Phenotype characterization is largely observational. In order to appreciate the range of phenotype penetrance, some categorical quantification is needed comparing density (or other parameter) of pigmentation in 3C ablated individuals, WT, and *tyr* mutants. This is not needed for all iterations yet quantification for the Cre mRNA injected 12 hpf HS or for the RPE KO would be useful and provide statistical rigor.

The reviewer is right that quantification of pigmentation loss in 3C *tyr* animals treated with different Cre sources was missing. In order to solve this we repeated the two experiments which the reviewer suggested and determined the degree of pigmentation in the retinal pigment epithelium. To this aim, 8 bit black and white images of embryos expressing no fluorescent protein, DsRed- and GFP-positive siblings as well as *tyrosinase* mutants (*tyr*^{-/-}) were taken at 50 hpf. Following the selection of the region of interest, covering the dimension of the developing eye, the defined area was applied manually to all images and the mean of the histogram was measured using FIJI (ImageJ). The description of the quantification has been added to methods, including a new supplementary figure explaining how the data were acquired (Supplementary Fig. 1). The measured data points are provided in the new supplementary tables 1 and 2. The quantifications have been added to figures 2 and 5 and their respective descriptions have been incorporated in the results (lines 128ff and 200ff). Importantly, in both cases the analysis strongly supports our conclusion that GFP-positive 3C *tyr* animals show a significant loss of pigmentation and are highly similar to unpigmented *tyrosinase* mutants.

2. As this is a F0 mutagenic strategy generating random genotypes – quantifying the rate of potentially functional in-frame mutations from your sequencing is needed. Discussing this in combination with a gross estimate of non-recombined loci would be helpful to better estimate the penetrance of 3C (IE non-recombined due to variable Cre mRNA expression or mosaicism of Cre driving transgenes). Demonstration of 3C effectiveness with a different gDNA target, alternative gene or target within *tyr* itself, would be ideal to highlight penetrance beyond an N=1, yet arguably beyond the scope of the current study.

We agree with the reviewer that the number of frameshift and in-frame mutations is important. We have added this information in figure 3, supplementary figure 3 as well as supplementary figure 7 and made the appropriate changes in the results (lines 151ff and 245ff). We also expanded the discussion to address this topic as the reviewer suggested (line 379ff).

With respect to different target sites, we are currently generating more 3C gene inactivation lines but the temporary shutdown due to the Covid19 pandemic and the tentative reopening of our institute has unfortunately severely affected the generation of the corresponding transgenic lines. We do however not expect any differences with other targets. The *tyrosinase* locus has been reported to be an “easy” target but only with respect to the readout of the mutant phenotype visible as a loss of pigmentation. In contrast, the mutagenesis rate of the *tyr* target site is similar to other target sites as was shown in Jao et al., 2013 (in *PNAS* 110, 13904-13909). In this aspect, determination of highly efficient target sites will be key for the generation of future 3C gene inactivation lines as we point out in the discussion (line 353ff).

3. Can the authors address the lack of leakiness from the pHSP1 promoter? The basal promoter activity of pHSP in zebrafish is well established and this promoter is used as a minimal driver for enhancer traps screens. The authors should use a 3 primer test, similar to Sup 5C, with an additional primer in dsRed and show a low and high exposure gel, in order to absolutely confirm the complete absence of non-induced gene disruption, as the authors state. This is important to rigorously address 3C stability and therefore the reliability of control experiments. If the authors confirm that their construct is absolutely not leaky, they should comment as to the cause – insertion position, which would be a great interest as well.

This must be a misunderstanding. We did not want to imply that the *hsp70l* promoter in our 3C construct is perfectly tight but rather that the potential transcriptional basal levels do not translate into enough Cas9-GFP protein that is capable to elicit mutagenesis of the target site. If this would be the case, we would expect that 3C *tyr* animals subjected to Cre mRNA injection but no heat treatment show either some pigmentation defects (indicating even biallelic conversion) or at least a reduction in the frequency of the parental strands in the sequencing analysis. (Analog to a robust site-specific recombination with the use of a *hsp70l*:Cre transgene which we described in Hans et al., 2011). However, we do not observe anything like this in the case of our recombined *hsp70l*:Cas9-GFP transgene. With respect to pigmentation, 3C *tyr* animals subjected to Cre mRNA injection but no heat treatment are indistinguishable from non-transgenic wild-type controls at larval and adult stages. With respect to sequencing, the frequency of the parental strands in 3C *tyr* animals subjected to Cre mRNA injection but no heat treatment is 90.9%. This is in the same range as the controls with 92.54% in non-transgenic animals as well as 90.87% and 91.47% in non-recombined 3C *tyr* animals with and without heat treatment, respectively. We hence concluded that the basal leakiness of the *hsp70l* promoter driving Cas9-GFP does not result in the production of enough gene product eliciting non-conditional mutagenesis. To make this clearer, we changed the text in the results (line 217ff and 259ff).

We did not execute the experiment that the reviewer suggested. A PCR with three primers would not help us to answer the question if basal leakiness of the *hsp70l* promoter results in non-conditional mutagenesis. It would rather only answer if the 3C construct undergoes site-specific recombination in all cells or if the 3C construct remains unrecombined in some cells. Although the suggested experiment would provide additional evidence, this information is actually already given in the FAC-sorting experiment that revealed the presence of GFP-negative cells in GFP-positive embryos and is provided in the discussion (line 304ff).

Minor

1. The authors assert that 3C is scalable and viable for multi target knockout, yet this is not currently demonstrated. Therefore, in the abstract this should not be a definitive statement, yet I agree an apt point for discussion.

We agree with the reviewer. Because reviewer 1 raised additional issues, we significantly revised large parts of the abstract (line 23ff). We also rephrased the statements in the introduction and discussion (lines 91ff and 349ff).

2. The authors should discuss their system in relationship to the heat shock induced CRISPR ablation recently published: Wu YC, Wang IJ. Heat-shock-induced tyrosinase gene ablation with CRISPR in zebrafish. *Mol Genet Genomics*. 2020 Jul;295(4):911-922. doi: 10.1007/s00438-020-01681-x. Epub 2020 May 4. PMID: 32367255.

We followed the suggestion of the reviewer and added this information in the discussion (line 318ff).

3. Line 210. Figure 4A may not be the correct figure reference.

We corrected the typo.

4. Including pictures of *tyr* mutants in some figures would be helpful to highlight knockout phenotype.

This is a very good suggestion and we added images of constitutive *tyr* mutants in figure 2.

5. Is there a proposed reason the parent allele percentages are dramatically divergent (notable Sup 6F)?

We actually have no explanation for the observed phenomenon. We also noticed the significant difference in the parental strand percentages. As the reviewer states and as it was shown in the former supplementary figure 6f, the presence of the parental strand with the PAM sequence "AGG" drops from 83.94% in controls to 20.64% in GFP-positive embryos. The presence of the parental strand with the PAM sequence "TGG" drops more drastically from 8.6% in controls to 0.85% in GFP-positive embryos. This indicates that the target site with the PAM sequence "TGG" is recognized and modified by Cas9 with a higher efficiency. However, in the first experiment using FAC-sorted cells (shown in Fig. 3) the presence of the parental strand with the PAM sequence "AGG" drops from 48.26% in control cells to 0.48% in GFP-positive cells whereas the presence of the parental strand with the PAM sequence "TGG" only drops from 38.46% in control cells to 7.15% in GFP-positive cells. Here, the conclusion is that the target site with the PAM sequence "AGG" is recognized and modified by Cas9 with a higher efficiency. Hence, we obtained opposite results. We also conducted a literature research but retrieved only one publication reporting very subtle differences in the PAM preference of Cas9 protein in human cells (Tang et al., 2019 in *Cell Regeneration* 8, 44-50). Taken together, the data of a potential PAM preference was inconclusive and we were unable to provide an interpretation which is why we left this topic undiscussed.

6. Some discussion of studies that used a F0 CRISPR mediated mutagenic strategy(1–3) would be helpful and highlight the applicability of this approach.

We agree with the reviewer. We made the appropriate changes in the discussion (line 320ff) and added the relevant references.

7. Can the authors envision scenarios where 3C would not be ideal – highly restricted patterns of expression etc? Adding this to the discussion would be valuable insight to labs interested in applying 3C.

As we point out in the discussion, our 3C mutagenesis system provides several clear advantages over the production of floxed alleles. However, 3C also has some limitations that can currently only be addressed with different approaches. Among them, we describe CRISPR-Switch in the discussion (line 412ff). CRISPR-Switch controls the production of gRNAs in a Cre-dependent manner and allows sequential gene editing of two loci in a temporal order. However, we believe that sequential mutagenesis of two genes will rather represent a special case. We envision that simultaneous gene inactivation of two or more genes, which should be possible with 3C, will be a broader application, in particular in zebrafish with its partial genome duplication.

8. Line 345 – can a reference be provided for low recombination tissue?

We added the reference (line 368ff).

9. Can the authors provide a rate for F1 transmission following Cre recombination?

We included this information in the results (254ff).

10. The authors discuss using phi landing sites for 3C constructs. Out of curiosity, do the authors think using similar techniques flanking the gRNA locus specifically would be useful for rapid gene target swapping?

Based on our experience with PhiC31 integrase, we do not think that rapid gene target swapping will be possible due to limited efficiency. In our hands, PhiC31-mediated integration results in a 30% germline transmission rate, which is consistent with previous reports (Mosimann et al., 2013). This number is sufficient for the generation of new lines but it is not high enough to perform gene target swapping in F0 animals. Hence, we believe that the use of landing site lines is beneficial but will require the establishment of new lines.

11. Addressing CRISPR knockout mediated compensation(4) in the discussion would be immensely helpful and an important issue to consider when applying 3C to different genetic targets.

This is indeed an important point, which we did not address in the first submission. We corrected this in the revised version, made the appropriate changes to the text (line 324ff) and added the relevant references.

Reviewers' Comments:

Reviewer #1:

Remarks to the Author:

The authors have addressed all comments from my original review appropriately. There is one unclear sentence in the new discussion. The sentence starting line 324 "In this context it is feasible that the phenomenon of genetic compensation is not or only slightly induced in this strategy." is not clear. Should it be "It is conceivable that the phenomenon of genetic compensation..."?

Reviewer #2:

Remarks to the Author:

The authors have thoroughly addressed my concerns, adding new experiments, representative images, and analysis. This work will be an exciting contribution to the scientific community.

Reviewer #1 (Remarks to the Author):

The authors have addressed all comments from my original review appropriately. There is one unclear sentence in the new discussion. The sentence starting line 324 “In this context it is feasible that the phenomenon of genetic compensation is not or only slightly induced in this strategy.” is not clear. Should it be “It is conceivable that the phenomenon of genetic compensation...”?

We agree with the reviewer that the sentence is not entirely clear. We hence followed the suggestion and changed it accordingly.

Reviewer #2 (Remarks to the Author):

The authors have thoroughly addressed my concerns, adding new experiments, representative images, and analysis. This work will be an exciting contribution to the scientific community.

We appreciate the statement and thank the reviewer for the comments and suggestions.